# Environmental and anthropic factors influencing *Aedes aegypti* and *Aedes albopictus* (Diptera: Culicidae), with emphasis on natural infection and dissemination: Implications for an emerging vector in Colombia

**Juan S. Mantilla-Granados**[1]*, **Karol Montilla-López**[1], **Diana Sarmiento-Senior**[1], **Elver Chapal-Arcos**[1], **Myriam Lucía Velandia-Romero**[2], **Eliana Calvo**[2], **Carlos Andrés Morales**[3], **Jaime E. Castellanos**[2]*

1 Universidad El Bosque, Vicerrectoría de Investigaciones, Saneamiento Ecológico, Salud y Medio Ambiente, Bogotá, Colombia, 2 Universidad El Bosque, Vicerrectoría de Investigaciones, Grupo de Virología, Bogotá, Colombia, 3 Secretaría de Salud del Cauca, Laboratorio de Salud Pública, Laboratorio de Entomología, Popayán, Colombia

* jmantillag@unbosque.edu.co (JSMG); castellanosjaime@unbosque.edu.co (JEC)

## Abstract

### Background

Viruses such as the dengue virus (DENV), Zika virus (ZIKV), and chikungunya virus (CHIKV) pose major threats to human health, causing endemic, emerging, and reemerging diseases. These arboviruses have complex life cycles involving *Aedes* mosquitoes, driven by environmental, ecological, socioeconomic, and cultural factors. In Colombia, *Aedes aegypti* is the primary vector, but *Aedes albopictus* is expanding across the country. Understanding the unique characteristics of each species is crucial for managing arbovirus spread, particularly in areas where they coexist.

### Methodology/Principal findings

We conducted an entomological survey of *Ae. aegypti* and *Ae. albopictus* (larvae, pupae, and adults) in urban and rural areas of four municipalities across different elevations (200–2200 meters above sea level (masl)) in Colombia. Household conditions and knowledge of DENV were assessed through interviews. Female *Ae. albopictus* were tested individually for arbovirus RNA, while *Ae. aegypti* were tested in pools (as the accepted primary arbovirus vector in the country). Both species were found up to 2100 masl. *Ae. aegypti* comprised 78% of the immature forms collected, while *Ae. albopictus* made up 22%. Larvae from both species coexisted in common artificial breeding sites in urban and rural areas, with no evidence of competition. *Ae. albopictus* preferred rural areas, lower elevations (<1500 masl), high precipitation (>270 mm), and lowest household conditions, while *Ae.*

**Data availability statement:** All relevant data are in the manuscript and its Supporting Information files.

**Funding:** This project was funded by Minciencias and Universidad El Bosque contracts 891-2019, 489-2021 to JSM, DSS, MLV, CAM, and JEC, and call 909 de 2021 to JSM and DSS. The funding sources had no role in study design, data collection, analysis, or preparation of the manuscript.

**Competing interests:** The authors have declared that no competing interests exist.

*aegypti* was more abundant in urban areas, intradomicile environments, and areas with moderate precipitation (100–400 mm). Potential female-human contact was higher for *Ae. aegypti* (0.02–0.22 females per person), particularly in urban areas, while *Ae. albopictus* exhibited lower female per person: 0.001–0.08, with the highest values in rural Patía. Natural infections of DENV (12.4%) and CHIKV (12.4%) were found in *Ae. aegypti*, while *Ae. albopictus* showed CHIKV (41.2%) and DENV (23%) infections, with virus dissemination to the legs and salivary glands.

## Conclusions/significance

Integrating household conditions and community knowledge with environmental data can enhance predictive models for the eco-epidemiological characterization of Aedes-borne viruses, especially in areas where two vector species with distinct ecological characteristics coexist. Our findings highlight the need to consider *Ae. albopictus* as a potentially significant arbovirus vector in Colombia, especially given the presence of arboviruses in its salivary glands, its use of artificial breeding sites, its biting risk inside homes, and its differing ecological preferences and seasonal associations compared to *Ae. aegypti*.

## Author summary

In this study, we investigated the ecological and epidemiological dynamics of *Aedes aegypti* and *Aedes albopictus* and their natural infection with DENV, ZIKV, and CHIKV in Colombia. An entomological survey conducted across four municipalities revealed distinct environmental and human-related factors influencing the distribution and abundance of these species. *Ae. aegypti* was more abundant in urban environments, favoring areas with lowest precipitation and a range of household conditions, while *Ae. albopictus* was more common in rural areas with higher precipitation and poorer household conditions. Both species were naturally infected with DENV and CHIKV, with *Ae. albopictus* showing the ability to disseminate, as indicated by their presence in the legs and salivary glands. Our findings underscore the importance of understanding species-specific ecological characteristics and incorporating social characteristics, particularly in regions where both species coexist and contribute to arbovirus transmission.

## Introduction

Arboviruses account for > 17% of infectious diseases worldwide, with a high proportion of emerging and zoonotic pathogens concentrated in tropical and subtropical regions [1]. Dengue virus (DENV) is the most significant arboviral threat in the Americas, causing the highest morbidity and mortality, and is endemic across tropics and subtropics. In 2024, the Pan American Health Organization reported a 235% increase in DENV cases compared to 2023, and a 431% increase over the 5-year average [2]. The introduction of chikungunya virus (CHIKV) and Zika virus (ZIKV) to the Americas in 2013 and 2015, respectively, has also raised major public health concerns owing to their association with chronic diseases, neurological effects, and congenital infections [3–5]. Despite vector control efforts, the incidence of these arboviruses is increasing in the region, likely driven by climate change and unplanned urbanization [6]. Shifts in incidence may also result from changes in the epidemiological landscape due to the introduction of new competent vectors.

The primary vector for these arboviruses is the African mosquito, *Aedes aegypti* (L.), which has long been established in the Americas. It breeds in artificial containers, mainly in urban areas; however, it can also colonize rural settings[7]. The Asian tiger mosquito *Aedes albopictus* (Skuse) is a highly adaptable species that can use both natural and artificial larval habitats, allowing it to thrive in tropical and temperate areas, and at altitudes higher than that for *Ae. aegypti* [8–10]. *Ae. albopictus*, is a secondary or even primary vector for DENV, ZIKV, and particularly CHIKV, depending on genetic factors that influences its vector competence and interactions with the local microbiota [11,12].

Field-collected mosquitoes can be screened for the presence of arbovirus in the whole body, abdomen, and midgut to assess natural infection, which may reflect an ingested or disseminated virus. Virus detection in the legs or organs outside the midgut indicates dissemination, whereas infection in the salivary glands or saliva is associated with transmission potential [11,13].

*Ae. aegypti* and *Ae. albopictus* occupy similar ecological niches and may compete, with *Ae. albopictus* occasionally displacing *Ae. aegypti* [14–16]. As invasive species, both are well adapted to human-modified environments, and their presence can be influenced by climatic and social factors that may sustain arboviral transmission over extended periods [17]. While both vectors are present in the Americas, *Ae. albopictus* is considered a secondary vector due to its limited distribution, relatively low population densities, and reduced vector competence reported in American populations from the United States and Brazil [18,19]. However, aspects of vector competence, such as the capacity for arbovirus dissemination in field-collected *Ae. albopictus* populations from endemic countries like Colombia, remain largely understudied. This knowledge gap is particularly critical, because environmental factors, local socio-economic conditions, and mosquito genetic variability could influence its ability to transmit arboviruses such as DENV, CHIKV, and ZIKV.

In Colombia, *Ae. albopictus* was first reported in 1998 and has since spread to 22 departments of the country. This mosquito is naturally infected by DENV, ZIKV, and CHIKV [20–25] with some studies demonstrating its ability to colonize major urban areas [26]. However, the role of this species as a vector in Colombia, its relationship with *Ae. aegypti*, and the climatic and anthropogenic factors associated with their distribution are poorly understood.

The Cauca Department is an ideal study area for these issues, as both *Aedes* species have been widely reported there. The region experiences a high incidence of dengue, and various dengue control efforts using educational, physical, and chemical approaches have been implemented. Therefore, we aimed to investigate the presence of *Ae. aegypti* and *Ae. albopictus* in relation to climatic and anthropogenic determinants in this setting, their natural infection status, and arbovirus dissemination in *Ae. albopictus*. Overall, our results revealed distinct ecological patterns for *Ae. aegypti* and *Ae. albopictus* populations across different environmental and anthropic conditions in the surveyed Colombian municipalities. Arbovirus infections, including DENV and CHIKV, were detected in both species, with notable virus dissemination and salivary glands infections in *Ae. albopictus*, suggesting its role as an arbovirus vector.

## Methods

### Ethical considerations

Before data collection, written informed consent was obtained from all participants. This study was approved by the Institutional Ethics Committee for Research at Universidad El Bosque, Bogotá, Colombia (act number: 013–2021). Mosquito sampling was authorized by the National Authority of Environmental Licenses (resolution number: 01470, November 17, 2017).

## Study area

This study was conducted in the urban and rural areas of four municipalities of the Cauca Department, selected for sampling at certain altitudinal ranges: Piamonte (0–600 m above sea level (masl)), Patía (600–1200 masl), Piendamó (1200–1800 masl), and Popayán (1800–2200 masl) (Fig 1). According to Colombia's national classification for dengue transmission (2021–2023), Piamonte, Piendamó, and Popayán are categorized as low transmission risk areas, while Patía is classified as having a medium transmission risk [27].

## Sample size

This cross-sectional study used the household as the unit of analysis, with the sample size determined using projected population data from the national census (https://www.dane.gov.co/) and calculated with SSPropor tool (http://www.openepi.com/). Households were selected based on their altitude and prior reports of dengue cases or mosquito presence, as provided by the health authorities of each municipality. Each household was visited twice: once during the dry season (July–October 2021) and once during the wet season (March–May 2022) for mosquito collection.

## Mosquito collection

Mosquito sampling was conducted between July–October 2021 and March–May 2022. Immature mosquitoes were collected directly from water containers in urban and rural settlements. The containers were categorized as either artificial or natural. Artificial containers were defined as man-made objects capable of holding water, included laundry sinks, ground tanks (storage tanks with a capacity >2000 L placed on the ground), tires, buckets, and drain sewers. Less common artificial containers were classified under the category 'others.' Natural containers, defined as naturally occurring structures or formations that collect water, included bamboo, aquatic plants, tree holes, and plant axils. This classification was applied to containers found in three spatial areas: intradomicile (inside the household), peridomicile (0–10 m from the household), and extradomicile (>10 m from the household) [28].

Sunlight exposure at breeding places was also recorded as uncovered (direct sunlight), partially covered or covered (completely under the shadow). Pupae were allowed to emerge adults. Adult resting mosquitoes were captured using a Prokopack aspirator [29], for 15 min in intradomicile and peridomicile areas, while those in extradomicile areas were collected using entomological nets when observed. Collection sites were georeferenced using Garmin GPS 62s (Garmin International Inc., Olathe, KS, USA).

## Mosquito identification

Immature and adult mosquitoes were taxonomically identified in the field using the morphological characteristics previously described [30,31] for each species, using a stereo microscope Zeiss Stemi DV4 (Jena, Germany). The identified *Ae. aegypti* specimens were pooled by life stage, sex, date, and location, and preserved in RNAlater Stabilization Solution (Ambion, AM7020) (Thermo Fisher Scientific, Waltham, Massachusetts, U.S.A.). Since the dissemination capacity of Colombian *Ae. albopictus* populations has not been evaluated, females were dissected into the salivary glands, midgut, legs, and thorax, and each part was preserved individually in RNAlater. Males and immature mosquitoes of both species were processed similarly to *Ae. aegypti*.

## Entomological index

The following indices were calculated to assess mosquito infestation levels: the percentage of key positives breeding places, calculated as the number of positive containers in each category

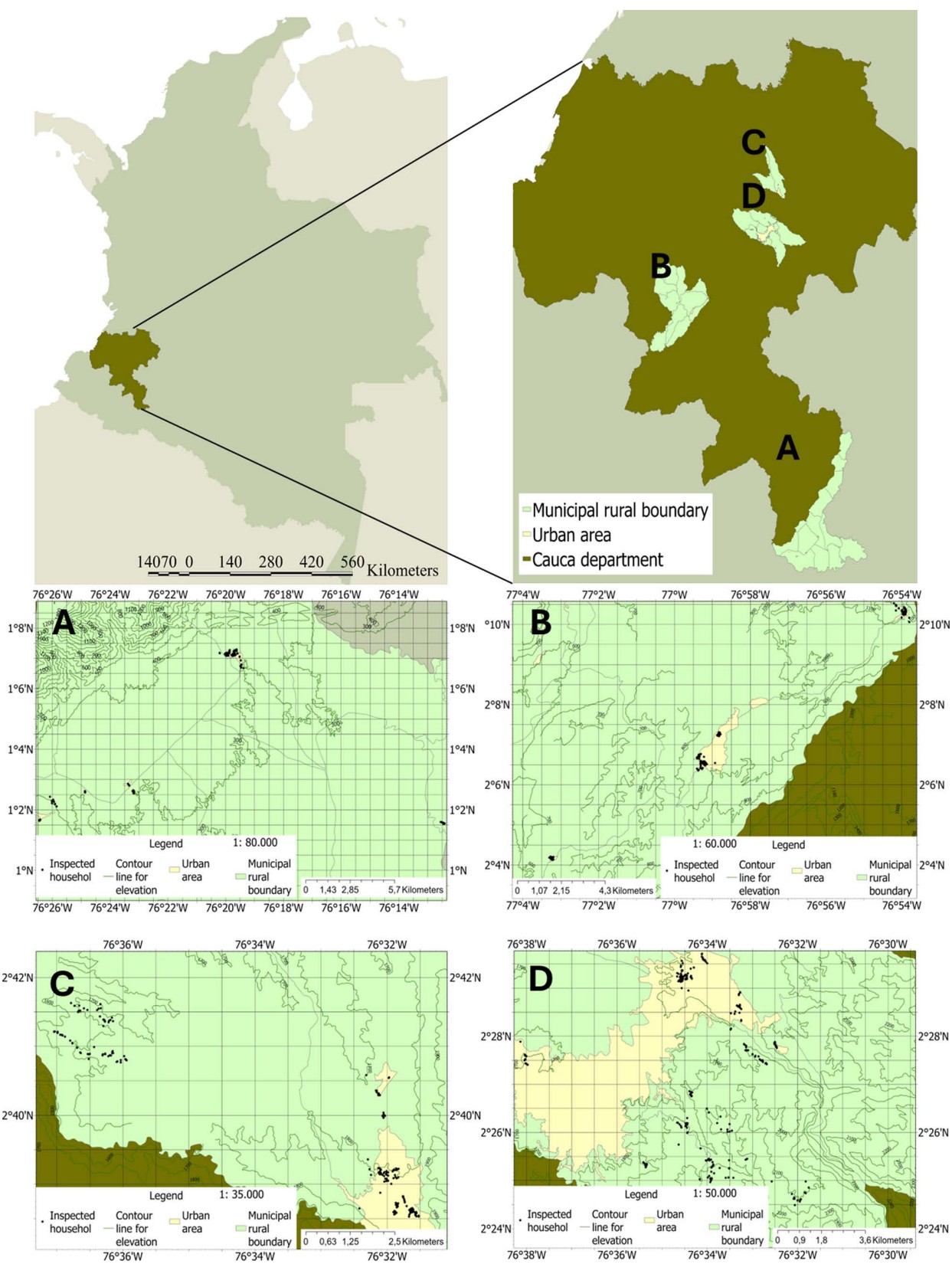

**Fig 1. Map for the sampled localities in Colombian Map showing the localization of Cauca department in olive and zoom of the sampled municiplaities.** A–D Municipality detailed maps, with the location of the urban (light yellow) and rural areas (light green), and contour lines

for elevations. A. Piamonte. B. Patía. C. Piendamó. D. Popayán. Black dots indicate the localization of sampled households. Base layer maps for country departments, urban and rural borders were obtained from open files of: Departamento Administrativo Nacional de Estadística - DANE: www.dane.gov.co at geoportal: https://geoportal.dane.gov.co/servicios/descarga-y-metadatos/datos-geoestadisticos/, allowed to be used under Creative Commons Attribution 4.0 International (https://geoportal.dane.gov.co/acerca-del-geoportal/licencia-y-condiciones-de-uso/#gsc.tab=0), and contour line for elevations were calculated in ArcGis Pro, from digital elevation models downloaded at: http://www.usgs.gov

divided by the total number of positive containers for each species; the Breteau index, calculated as the number of positive water-holding containers per 100 houses divided by the total number of inspected houses; and the pupae-per-person index, which was determined as the number of collected pupae divided by the number of household inhabitants (as reported by interviewed residents) [32]. For adult mosquitoes, we calculated the Female infestation rate – Number of female mosquitoes collected per household during a 15-minute Prokopack aspirator sampling in intradomicile areas. Additionally, the females-per-person index was calculated by dividing the female infestation rate per household by the number of household inhabitants. Mosquito biting risk was evaluated as the potential contact between mosquitoes and humans, based on the results of pupae-per-person and female-per-person indexes. The total number of collected larvae and pupae of each species per house was also analyzed.

## Environmental variables

The average, minimum, and maximum temperature values, along with total precipitation for the collection periods, were obtained from WorldClim data [32]. Geographic coordinates and elevations recorded in the field were projected and linked to environmental variables using ArcGIS Pro v.3.3.

## Anthropogenic variables

Residents of each sampled household were interviewed about various sociodemographic and economic characteristics, including access to running water (and its continuity), water storage and rainwater harvesting practices, trash disposal services and methods (burying or burning), and primary construction materials (concrete, wood, or other, such as shade mesh walls).

Additionally, general knowledge about dengue transmission, the most common arbovirus in the study area, was assessed. Interview questions covered prior exposure to dengue-related information, understanding of transmission mechanisms, identification of at-risk populations, and recognition of correct and incorrect *Aedes* breeding sites. To prevent confusion with other mosquito species, local health authorities were consulted regarding commonly used names for the adult and immature stages of dengue vectors. Residents were also asked about the "white-striped mosquito" to distinguish it from *Culex* species.

The collected data were categorized as binary (0/1) variables and analyzed using principal component analysis (PCA) with the FactoMineR library in R v.4.3.3. To enhance data variance explanation, we retained the most representative variables for the first two principal components: The first component ("house conditions") corresponded to household infrastructure similar to the previous described environmental capital [33]. Positive scores were associated with running water, sewerage, and concrete-based construction, whereas negative scores indicated absence of these services, wooden construction, and waste burning practices. Zero scores represented intermediate conditions, where households had some but not all public services.

The second component ("knowledge") captured awareness of dengue and its vector (S1 Fig). Retained variables included prior exposure to dengue-related information, recognition of breeding sites, and identifying mosquito bites as the transmission mode. Higher scores

indicated accurate knowledge, lower scores reflected misinformation, and zero scores represented a lack of awareness. Household coordinates along these PCA components were used as response variables in further statistical analyses.

## Statistical analysis

To analyze larvae abundance of each species in relation to breeding place characteristics and abundance of the other species. The analysis was performed by using a generalized linear model (GLM) with poisson distribution using function 'glm' from MASS [34] package in R program version 4.3.3. For female infestation, the number of larvae of each species was also analyzed against environmental variables, housing conditions and knowledge were separately evaluated because of collinearity. The presence of multicollinearity was assessed using the generalized variance inflation factor $GVIF^{(1/(2 \times Df))}$, with the 'car' package [35] in R 4.3.3. The data exhibited nonlinear patterns with the response variables; therefore, generalized additive models (GAMs) with negative binomial distributions were implemented using mgcv package[36] in R v.4.3.3 Cubic regression spline smoothing was applied with the k parameter manually adjusted for each response variable.

## RNA extraction and amplification

RNA was extracted from mosquito samples using a Viral Nucleic Acid Extraction Kit II (IBI Scientific, Chavenelle, Dubuque, U.S.A.), according to the manufacturer's protocol, and quantified using a RNA samples were quantified using the Nanodrop system NanoPhotometer NP80 (Implen, Westlake Village, California, U.S.A.). Viral RNA was reverse-transcribed and amplified by multiplex nested reverse transcriptase–polymerase chain reaction using a Luna Universal kit One-Step RT-PCR (Thermo Fisher Scientific, Waltham, Massachusetts, U.S.A.) and primers for DENV, CHIKV, and ZIKV, as previously described [37]. Culture-harvested viruses were used as positive controls and nuclease free water as negative control of reaction. Amplified products were separated on 2% agarose gels and visualized by staining with ethidium bromide. For *Ae. albopictus* samples, abdominal or midgut amplifications were recorded as natural infections, whereas leg/thorax or salivary gland amplifications indicated viral dissemination, the percentage of positive pools or individuals (complete or dissected) was calculate, in the case of pools the minimum infection rate (MIR) was calculated as the number of positive pools/ total specimens tested x 1000.

## Results

In total, 904 households were sampled across urban (n = 566) and rural (n = 338) areas (S1 Table). Each household was originally planned for two visits: one during the dry season and one during the wet season. However, some households could not be revisited due to resident unavailability. The final sample sizes per municipality and season are listed in S1 Table.

## Anthropogenic variables

We selected the most representative variables from the first two principal components to enhance data variance explanation. For the first component, "house conditions," Positive scores (0 to 2.5) were associated with access to running water, sewerage, and concrete housing, while negative scores (-2.5 to 0) corresponded to houses lacking these services, using wood, and practicing waste burning. Zero scores indicated partial access to services. The second component, "knowledge," captured awareness of dengue and its vector (S1 Fig). Retained variables included prior exposure to dengue-related information, recognition of breeding sites, and understanding mosquito bites as the transmission mode. Positive scores (0–2) indicated

accurate knowledge, negative scores (-5–0) reflected misinformation, and zero scores corresponded to lack of awareness. Variables with minimal contribution were excluded, preserving the overall pattern. The first two components explained 30.3% and 22.7% of the data variance.

Urban and rural households differ significantly in house conditions (S1 Fig). In urban areas, 457 (80.7%) houses had access to running water, compared to 129 (38.1%) in rural areas. However, 50% (230) of urban households and 88.4% (114) of rural households reported irregular water supply. Water storage was practiced in 45% (255) of urban and 62.1% (210) of rural households. Burning waste was reported in 2.8% (16) of urban and 34.9% (118) of rural houses. Sewerage access was available in 68.4% (387) of urban and 20.4% (69) of rural houses. No significant differences were observed between urban and rural households in the knowledge component (S1 Fig).

## Immature stages

A total of 1,426 immature *Aedes* mosquitoes (larvae and pupae) were collected across the four municipalities, with 78% (1,112) *Ae. aegypti* (29% rural, 71% urban) and 22% (314) *Ae. albopictus* (64% rural, 36% urban) during the dry and wet seasons (Table 1). Both species were present in urban and rural areas, except in Piendamó, where *Ae. aegypti* was found only in urban settings. The Breteau index (BI) for *Ae. aegypti* was highest in Patía (18.2), followed by Piamonte (8.4), Popayán (1.6), and Piendamó (1.1). For *Ae. albopictus,* the highest BI was in Patía (4.8), followed by Piamonte (2.5), Popayán (1.5), and Piendamó (0.9).

**Table 1. Analysis of the number of larvae and female mosquitoes regarding urban location, water containers, and inhabitants, by a generalized linear model (GLM) with Poisson distribution.**

| Species | Variables | Larvae | | | Females | | |
|---|---|---|---|---|---|---|---|
| | | Estimate | Z.value | p-value | Estimate | Z.value | p-value |
| | Intercept | -11.8 | -0.06 | 0.94 | -2.2 | -10.01 | **<0.001** |
| | Number of *Ae. albopictus* larvae in the same container | 0.02 | 3.21 | **0.001** | | | |
| *Aedes aegypti* | Urban area (rural as reference) | 0.52 | 6.93 | **<0.001** | 0.79 | 5.39 | **<0.001** |
| | Wet season (Dry as reference) | -0.38 | -5.82 | **<0.001** | -0.90 | -6.43 | **<0.001** |
| | Number of water containers per household | | | | 0.22 | 2.55 | 0.01 |
| | Number of people living in the household | | | | -0.02 | -0.64 | 0.59 |
| | Sunlight exposition of the breeding place (Uncover) | 0.827 | 8.433 | **<0.001** | | | |
| | Sunlight exposition (Partial) | -0.41 | -2.66 | **0.008** | | | |
| | Location (Intradomicilie) | 1.08 | 2.81 | **0.005** | | | |
| | Location (Peridomicilie) | 0.23 | 0.5 | 0.54 | | | |
| | AIC | 6814 | | | 1720 | | |
| *Ae. albopictus* | Intercept | -1.79 | -0.003 | 0.997 | -3.24 | -7.94 | **<0.001** |
| | Number of *Ae. aegypti* larvae in the same container | 2.36 | 2.98 | **0.003** | | | |
| | Urban area (rural as reference) | -3.47 | -2.72 | **0.006** | -0.22 | -0.82 | 0.4 |
| | Wet season (Dry as reference) | 0.96 | 7.09 | **<0.001** | -1.59 | -4.26 | **<0.001** |
| | Number of water containers per household | | | | 0.55 | 3.91 | **<0.001** |
| | Number of people living in the household | | | | -0.08 | -1.07 | 0.28 |
| | Sunlight exposition of the breeding place (Uncover) | 9.78 | 0.52 | 0.59 | | | |
| | Sunlight exposition (Partial) | 1.40 | 8.03 | **<0.001** | | | |
| | Location (Intradomicilie) | -1.31 | -3.84 | **<0.001** | | | |
| | Location (Peridomicilie) | 6.33 | 2.03 | **0.04** | | | |
| | AIC | 2496 | | | 493.33 | | |

Both natural and artificial containers in intra- and extra-domiciliary areas, including parks, rivers, and forested zones, were examined (S1 Table). However, *Aedes* mosquitoes primarily used artificial containers as breeding sites (Fig 2 and S1 Table). *Ae. aegypti* most commonly bred in laundry sinks and buckets in both urban and rural areas. While *Ae. albopictus* preferred tires and laundry sinks in rural areas, while in urban settings, it mainly used buckets and laundry sinks. Both species were found in urban drain sewers, and *Ae. albopictus* occasionally used bamboo canes, though this was not a predominant breeding site (Fig 2).

Both *Ae. aegypti* and *Ae. albopictus* were found at elevations up to 2,100 masl (Popayán rural settlement). Generalized linear models revealed that larval abundance of *Ae. aegypti* was positively associated with the number of collected *Ae. albopictus* larvae in the same container, urban areas, uncovered and indoor breeding places. While it was negatively associated with wet season and partially covered breeding places (Table 1). For *Ae. albopictus*, larval numbers were significantly linked with the number of *Ae. aegypti* larvae collected in the same container, wet season, partial cover, and peridomiciliary breeding places; however, they were negatively affected by urban areas and indoor breeding places (Table 1).

The smooth curves from the GAM models indicated that *Ae. aegypti* larval numbers increased with minimum temperature above 13 °C, peaked at 18 °C, and declined at 21 °C. Precipitation between 100–400 mm was also associated with high number of *Ae. aegypti* larvae, which reached the maximum at approximately 250 mm and then decreased above

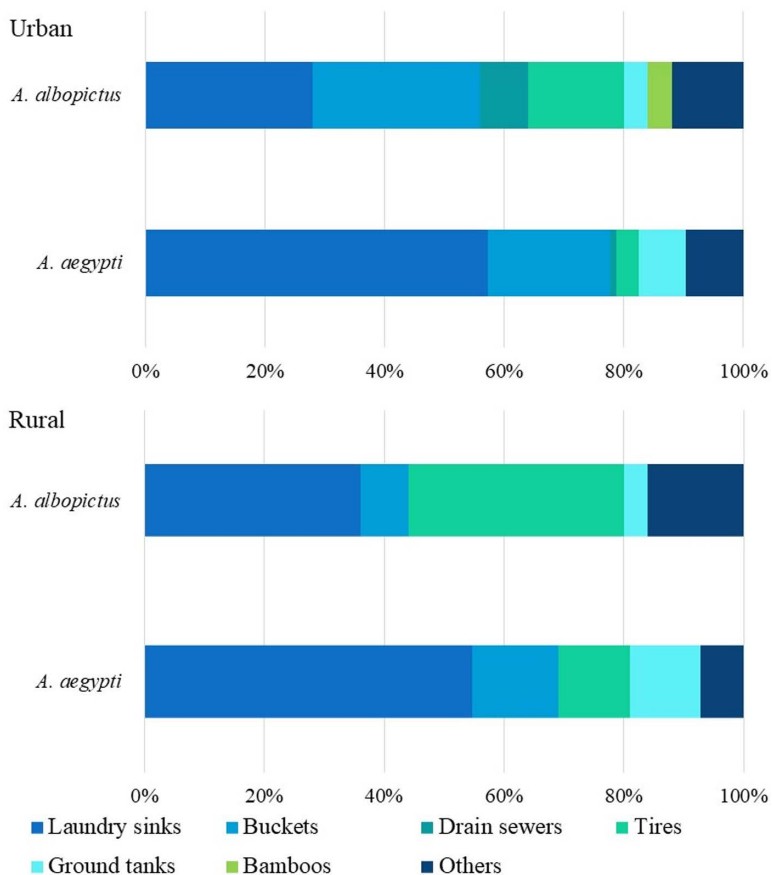

**Fig 2. Percentage of key breeding places productivity used in urban and rural areas by *Ae. aegypti* and *Ae. albopictus*.**

400 mm precipitation. Furthermore, *Ae. aegypti* larvae were more abundant at elevations between 100–1500 masl, peaking at 1000 masl (Fig 3). In contrast, *Ae. albopictus* larval numbers increased with minimum temperatures above 15 °C, and showed a strong positive relationship with precipitation above 270 mm, while elevations above 1500 masl reduced larval abundance (Fig 3).

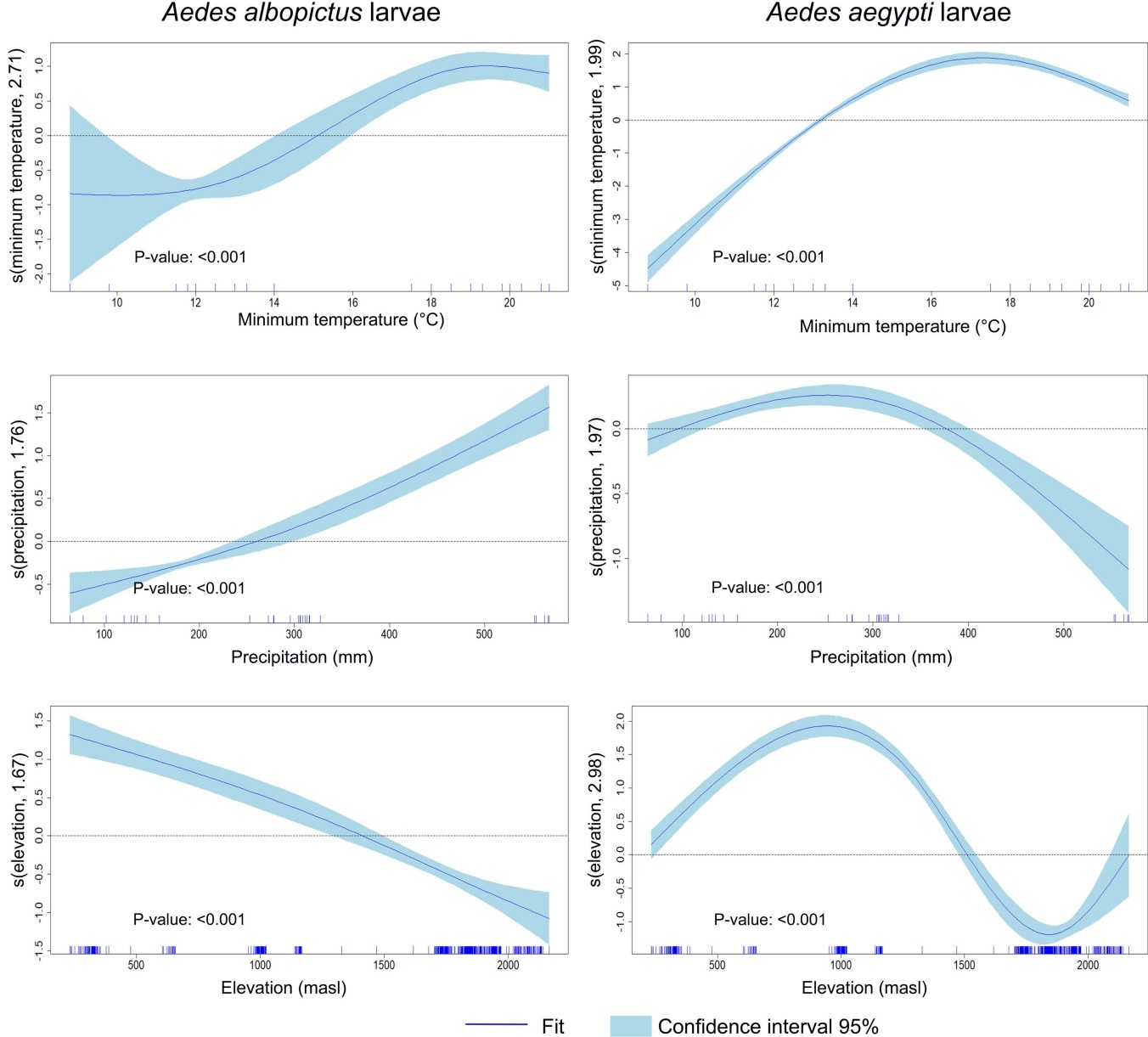

**Fig 3. Graphical representation of the estimated results of the GAM model analyzing the climatic factors regarding the number of larvae recollected per household.** Each panel of the GAM plot displays a smooth function of one predictor, illustrating how variations in that predictor influence the response variable (larvae number collected per each *Aedes* species), while the y-axis indicates the predictor's effect, interpreted as deviations from the average of the response variable, with zero indicating no effect and deviations indicating positive or negative influences. Each curve represents a non-parametric estimate of the predictor effect on the response, with shaded blue light areas indicating the 95% confidence interval, where a wider interval signifies more uncertainty. Additionally, the rug plot on the x-axis shows the observed data points' distribution, with clusters of tick marks highlighting areas of higher data concentration.

The abundance of *Ae. aegypti larvae* was positively associated with households at both the highest and lowest (negative values) ends of the smoothed terms for house conditions, as well as those with lack of awareness (Zero values) to misinformation (negative values) about dengue and its vector (Fig 4). In contrast, *Ae. albopictus* larval numbers increased in households

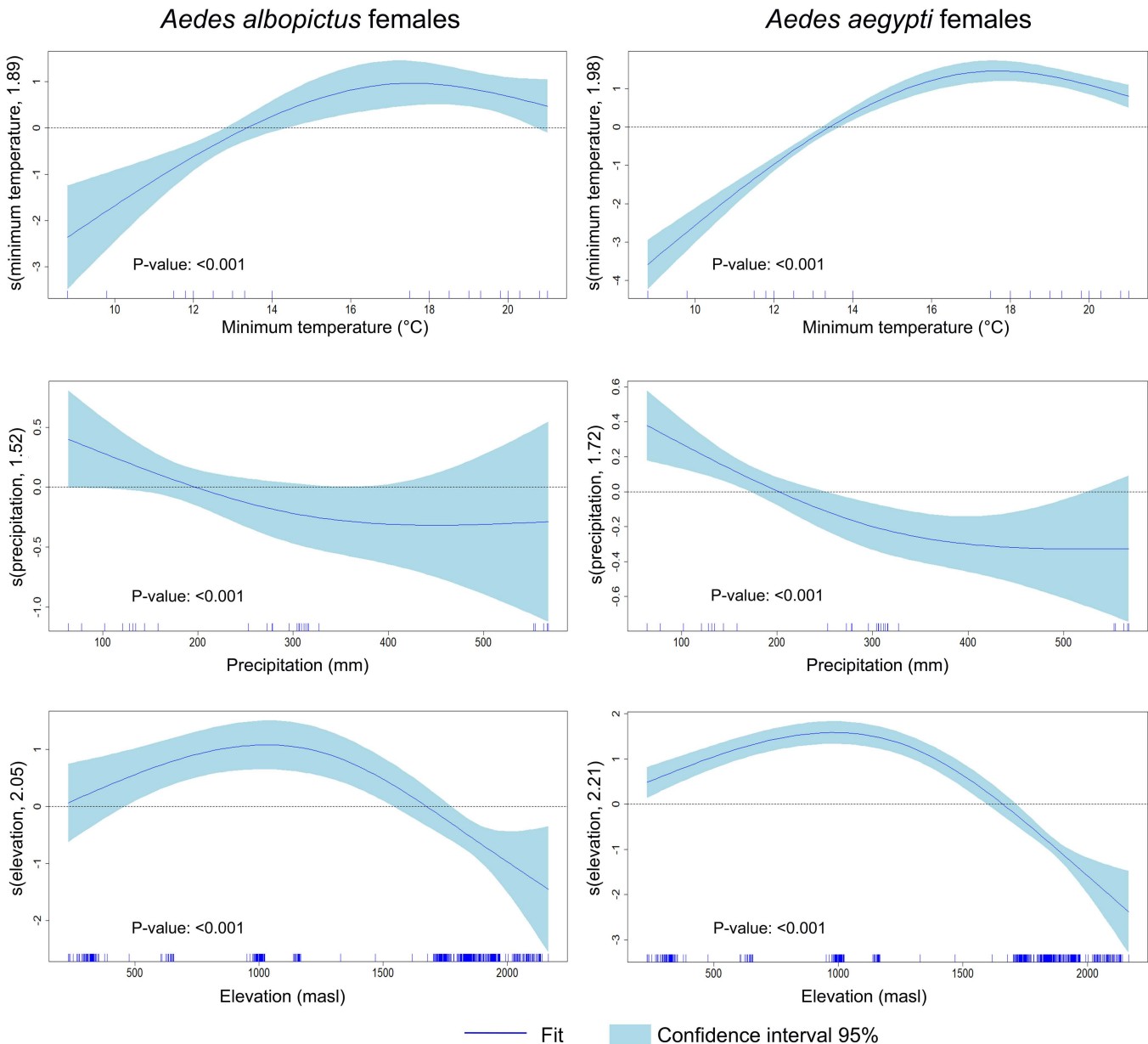

**Fig 4. Graphical representation of the estimated results of the GAM model analysis for the reduced anthropogenic variables termed as house conditions and knowledge, against the number of larvae and females recollected at each household.** Each panel of the GAM plot displays a smooth function of one predictor, illustrating how variations in that predictor influence the response variable (larvae or females number collected per each *Aedes* species), while the y-axis indicates the predictor's effect, interpreted as deviations from the average of the response variable, with zero indicating no effect and deviations indicating positive or negative influences. Each curve represents a non-parametric estimate of the predictor effect on the response, with shaded blue light areas indicating the 95% confidence interval, where a wider interval signifies more uncertainty. Additionally, the rug plot on the x-axis shows the observed data points' distribution, with clusters of tick marks highlighting areas of higher data concentration.

with negative values of house conditions primarily those constructed with wood and lacking public services (S1 Fig), the knowledge variable was similar as for *Ae. aegypti* (Fig 4).

## Adults

**Adult mosquitoes.** In total, 500 adult *Ae. aegypti* (110 from rural and 390 from urban areas) and 104 adult *Ae. albopictus* (49 from rural and 55 from urban areas) were collected. Among them, 238 *Ae. aegypti* and 86 *Ae. albopictus* were female. Notably, eight female and one male *Ae. albopictus* were found exclusively in extra-domicile environments in the rural areas of Patía and Popayán. Generalized linear models indicated that the presence of female *Ae. aegypti* was positively associated with urban areas, while female *Ae. albopictus* abundance was linked to the number of water containers per household (S1 Table).

Both species exhibited similar patterns for climatic variables. The number of adult female mosquitoes increased with the smoothed term for minimal temperatures above 15 °C, peaking approximately at 18–19 °C, and then declining at 21 °C. Precipitation > 200 mm negatively affected female abundance. Elevations between 100–1700 masl were associated with a relatively high number of females collected for both the *Aedes* species (Fig 5). Positive values for house conditions (the ones with running water, sewerage, and concrete housing) were positively associated with the number of female mosquitoes for both species; however, for *Ae. albopictus*, the number of females increased also in houses with negative house conditions values. Female *Ae. aegypti* was highly abundant in households where inhabitants lacked knowledge about dengue and its vectors, whereas female *Ae. albopictus* was more abundant in households with inaccurate knowledge about dengue and its vector (Fig 4).

## Risk of mosquito biting

Changes were observed in the mosquito-human contact among the study municipalities: in the case of *Ae. aegypti*, the density of females recollected per house inhabitant, was higher in both rural and urban areas of Patía; while the higher pupae per person index was observed in the urban areas of Piamonte (Fig 6). For *Ae. albopictus*, the females and pupae per person indexes were lower than that for *Ae. aegypti*, except in the rural areas in Popayán (Fig 6).

## Arbovirus infection

Natural arbovirus infections were detected in 23% (21/89) *Ae. aegypti* pools, with a minimum infection rate (MIR) of 94.38, Among these, 12.4% (11/89) were positive for DENV (MIR, 41.2) and 12.4% (11/89) for CHIKV (MIR, 49.4). For *Ae. albopictus,* as their abdomens or thorax were individually processed for the female mosquitoes, the infection rate as the percentage of infected abdomens was 61.7% (21/34), with 41.2% (14/34) by CHIKV and 23% (8/34) by DENV; no ZIKV infection was detected in females; however, ZIKV was found in male mosquitoes, indicating virus circulation (Table 2). For *Ae. aegypti*, the highest infection rate was found in the urban area of Piamonte (MIR, 171.4), followed by that of the urban areas of Patía (MIR, 115.8) and Popayán (MIR, 133.3). The most common natural infection in Patía was by CHIKV in both urban and rural areas, whereas DENV infection was predominant in Piamonte and Popayán. Among female *Ae. albopictus* from Patía, Piendamó, and Popayán, CHIKV was the most frequent infection, except in urban Popayán, where only one female tested positive for DENV (Table 2).

We detected evidence of DENV and CHIKV dissemination in mosquito tissues through RNA amplification in the legs and thorax of female *Ae. albopictus* (Table 3), indicating that the viruses successfully escaped the midgut barrier. Additionally, DENV serotype 2 and CHIKV

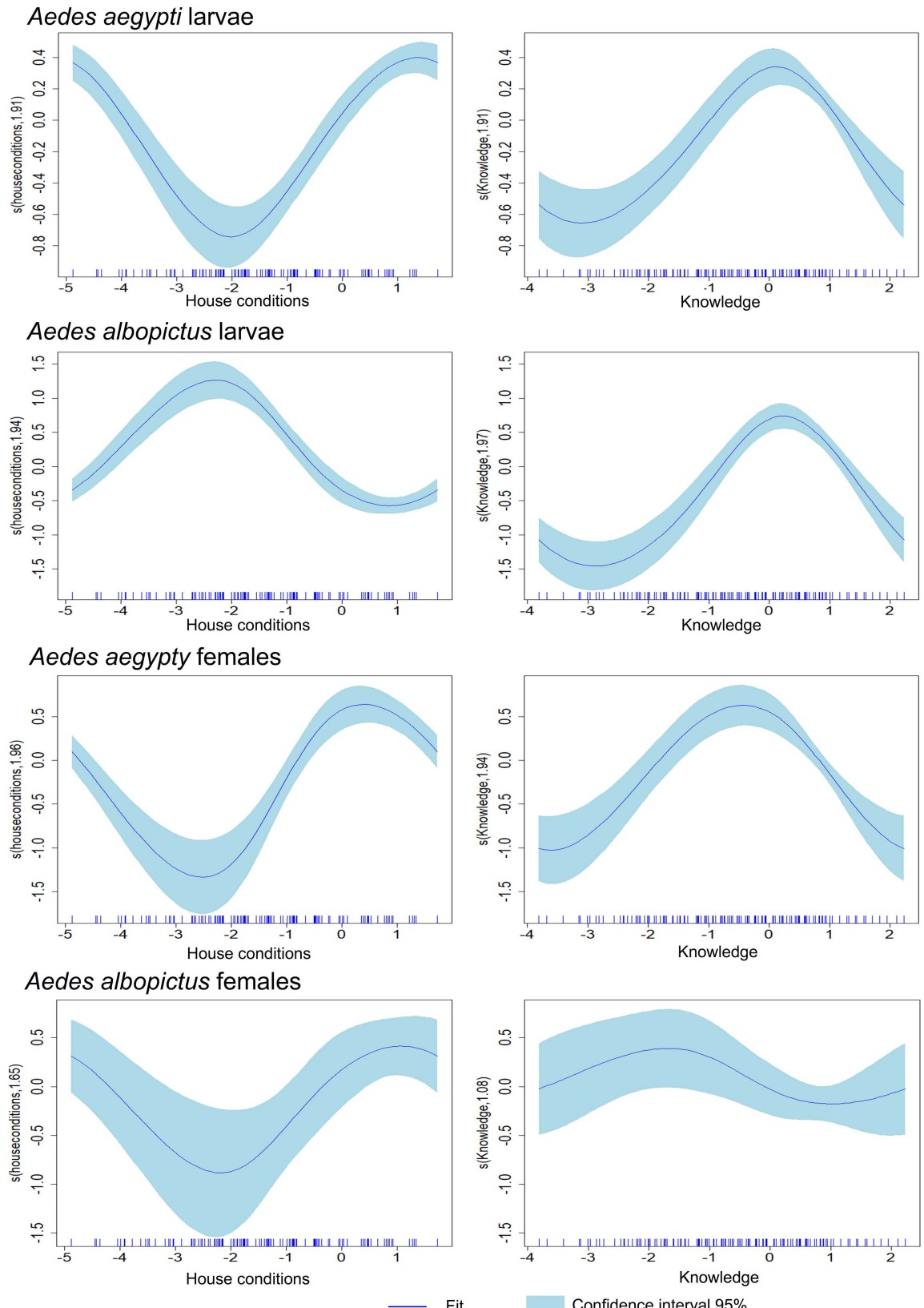

**Fig 5. Graphical representation of the estimated results of the GAM model analyzing the climatic factors regarding the number of females recollected per household.** Each panel of the GAM plot displays a smooth function of one predictor, illustrating how

variations in that predictor influence the response variable (females number collected per each *Aedes* species), while the y-axis indicates the predictor's effect, interpreted as deviations from the average of the response variable, with zero indicating no effect and deviations indicating positive or negative influences. Each curve represents a non-parametric estimate of the predictor effect on the response, with shaded blue light areas indicating the 95% confidence interval, where a wider interval signifies more uncertainty. Additionally, the rug plot on the x-axis shows the observed data points' distribution, with clusters of tick marks highlighting areas of higher data concentration.

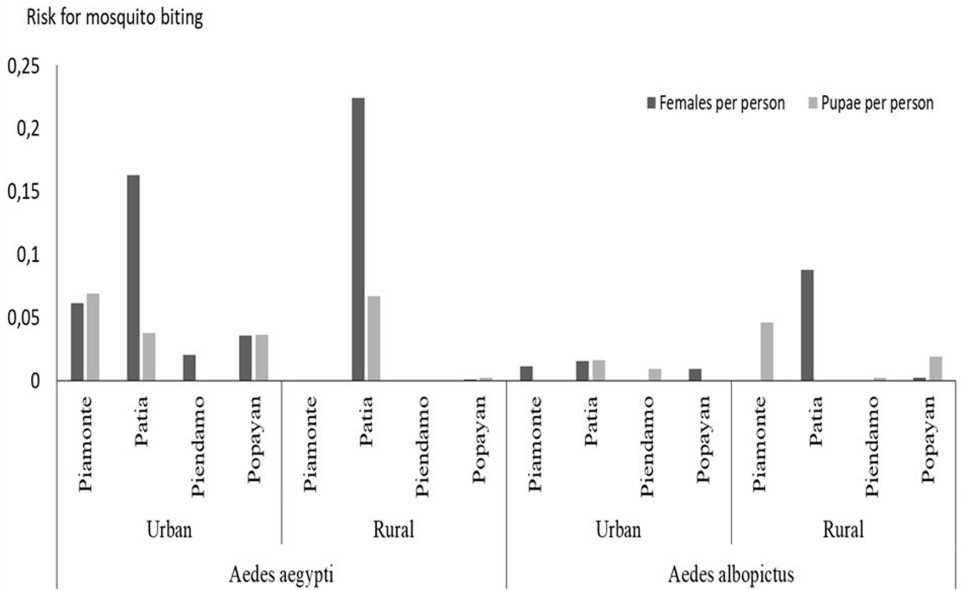

**Fig 6. Estimated rate for mosquito biting at each visited house, measure by the pupae per person index (number of pupae collected/number of house inhabitants) and by female per person (number of collected females/number of house inhabitants).** Rates were measured for the urban and rural areas of each municipality for both *Ae. aegypti* and *Ae. albopictus* species.

were detected in the salivary glands, providing further evidence that *Ae. albopictus* may act as a vector for both viruses.

## Discussion

Our integrative analysis of ecological, environmental, and sociodemographic factors, combined with molecular detection and mosquito dissections, provided detailed insights into the ecoepidemiology of Aedes-borne viruses in the study area. *Ae. aegypti* and *Ae. albopictus* were found colonizing areas up to 2,100 masl in Colombia, primarily utilizing artificial containers as shared breeding sites in both urban and rural habitats. The presence of these species was influenced by a combination of climatic and anthropogenic factors, with the latter analyzed through principal component analysis to derive key variables related to household conditions and knowledge. However, a larger sample size is needed to better assess the effects of seasonality and climatic variability on species distribution. Additionally, we detected DENV, ZIKV, and CHIKV circulating in mosquito vectors, and viral dissemination and salivary gland infection in female *Ae. albopictus*, suggesting its potential role as a vector in the study area. Nonetheless, further laboratory studies on its competence and vectorial capacity are essential for a clearer understanding.

 

**Table 2. DENV, ZIKV or CHIKV infection in *Ae. aegypti* pools and *Ae. albopictus* midguts or abdomens.**

| Species | Municipality | Area | Sex | CHIKV* | DENV* | ZIKV* | ARBOV* | Pro-cessed** |
|---|---|---|---|---|---|---|---|---|
| | **Piamonte** | urban | female | 2 (14.3) | 4 (28.6) | 0 | 6 (42.9) | 14 |
| | | | male | 0 | 1 (12.4) | 1(12.5) | 2 (25) | 8 |
| *Aedes aegypti* | **Patía** | Urban | female | 7(18.4) | 4 (10.5) | 0 | 11(28.9) | 38 |
| | | | male | 4 (11.1) | 1 (2.8) | 1 (2.8) | 6 (16.7) | 36 |
| | | rural | female | 2 (7.4) | 1 (3.7) | 0 | 2 (7.4) | 27 |
| | | | male | 0 | 0 | 0 | 0 | 15 |
| | **Piendamó** | urban | female | 0 | 0 | 0 | 0 | 4 |
| | | | male | 0 | 0 | 0 | 0 | 3 |
| | **Popayán** | urban | female | 0 | 2 (33.3) | 0 | 2 (33.3) | 6 |
| | | | male | 0 | 0 | 0 | 0 | 4 |
| | **Piamonte** | urban | female | 2 (14.3) | 4 (28.6) | 0 | 6 (42.9) | 14 |
| | | | male | 0 | 1 (12.4) | 1(12.5) | 2 (25) | 8 |
| *Aedes albopictus* | **Patía** | urban | female | 3 (60) | 1 (20) | 0 | 4 (80) | 5 |
| | | | male | 0 | 0 | 0 | 0 | 1 |
| | | rural | female | 7 (77.8) | 0 | 0 | 7 (77.8) | 9 |
| | | | male | 0 | 1 (20) | 0 | 1(20) | 5 |
| | **Piendamó** | rural | female | 1 (100) | 0 | 0 | 1 (100) | 1 |
| | | | male | 0 | 0 | 0 | 0 | 1 |
| | **Popayán** | urban | female | 0 | 2 (50) | 0 | 2 (50) | 4 |
| | | | male | 0 | 0 | 0 | 0 | 2 |
| | | rural | female | 1 (100) | 0 | 0 | 1 (100) | 1 |

*Number of pools (*Ae. aegypti*) or the number of midgut/abdomens (*Ae. albopictus*), in brackets the percentage is given.

**Total Pools (*Ae. aegypti*) or midgut/abdomen (*Ae. albopictus*)

**Table 3. Arbovirus molecular detection in legs/Torax or salivary glands from *Aedes albopictus* field collected females.**

| Municipality | Area | Legs/Thorax | | | | Salivary glands | | | |
|---|---|---|---|---|---|---|---|---|---|
| | | DENV | CHIKV | ZIKV | Processed | DENV | CHIKV | ZIKV | Processed |
| Piamonte | Urban | 0 | 0 | 0 | 5 | 0 | 0 | 0 | 1 |
| Patía | Urban | 3 (8.1) | 7(19) | 0 | 37 | 3(33) | 2(22) | 0 | 9 |
| | Rural | 3 (15) | 9(45) | 0 | 20 | 1(7) | 2(13) | 0 | 15 |
| Piendamó | Urban | 0 | 1(50) | 0 | 2 | 0 | 0 | 0 | 0 |
| Popayán | Urban | 0 | 2(50) | 0 | 4 | 0 | 0 | 0 | 1 |
| | Rural | 1(11) | 2(22) | 0 | 9 | 0 | 0 | 0 | 3 |
| Total | | 7(9) | 21(27) | 0 | 77 | 4(14) | 4(14) | 0 | 29 |

*Number of pooled legs or salivary glands from *Ae. albopictus* individuals, in brackets the percentage is given.

*Ae. albopictus* commonly inhabits peri- and extra-domicile environments [38]. However, our findings indicate that it is also present indoors, suggesting direct human contact. This species showed a strong capacity to colonize rural and urban habitats, as observed elsewhere [22,23]. Unlike studies highlighting natural breeding sites as key larval habitats for *Ae. albopictus* [14] our results suggest that artificial containers play a more significant role in its establishment. Both *Ae. aegypti* and *Ae. albopictus* were found up to 2100 masl, although *Ae. aegypti* has been documented at up to 2200 masl in Colombia [39], this is the highest reported elevation for *Ae. albopictus* in the country. This demonstrates the ability of mosquitoes to

  

colonize areas above 1800 masl as well as urban and rural habitats. However, their larval and adult numbers declined above 1500–1700 masl. A similar trend has been observed in Italy, where *Ae. albopictus* abundance decreases with elevation [40]. Additionally, the decline in mosquito abundance at higher elevations could be linked to improved house conditions, as Popayán—the highest elevation site—is a major city with better access to public services.

Previous studies using GAM models have reported a positive association between female *Ae. aegypti* infestation and minimum temperature, with collections peaking approximately at 18 °C [41], which is similar to our results. The presence of *Ae. albopictus* can be considerably predicted by combining climatic variables and neighborhood conditions [42], and we also found that the combination of climatic and anthropic factors provided a considerable understanding of the ecology of both species. Environmental capital related to house conditions improves the models for presence of *Aedes* mosquito, with households with medium environmental capital having relatively high immature collections [33]; however, this study did not distinguish between *Aedes* species. We found that the larval abundance of *Ae. albopictus* and *Ae. aegypti* changed associated with house conditions likely influenced by sociodemographic and behavioral factors [28]. Construction materials are associated with the presence of mosquitoes indoor, as non-concrete materials may facilitate females entry from outdoor breeding sites [43]. We also noticed that breeding places for *Ae. albopictus* were mainly found outside the house.

*Ae. albopictus* and *Ae. aegypti* show overlapping distributions and breeding sites, despite of the reports of competition and displacement [44]. In the present study, these species shared some breeding sites without the evidence for competing; however, *Ae. aegypti* was notably more abundant. The differential effects of climate and house conditions on their abundance are likely allowed *Ae. aegypti* and *Ae. albopictus* to avoid temporal overlapping in the study area by partitioning resources. This niche separation may prevent direct competition in terms of populations despite their cooccurrence and shared breeding sites.

Although laundry sinks the main breeding sites for both species are not directly affected by rainfall, smaller breeding sites, which are particularly important for *Ae. albopictus*, are influenced by the rainy season. Rainfall increases the availability of small water-holding containers, especially in households with certain structural and knowledge-related conditions, promoting mosquito presence in artificial water containers [45,46]. However, excessive rainfall can flush these containers, leading to larval mortality [47], which may explain the negative relationship between precipitation and *Ae. aegypti* larvae. In contrast, *Ae. albopictus* was associated with periods of high rainfall, possibly due to its egg-laying behavior and rapid hatching upon initial immersion [48], facilitating the co-occurrence of both species in small containers. Because *Ae. aegypti* eggs require multiple immersions to hatch, their ability to exploit these sites before rain flushes them is limited. This could favor *Ae. albopictus* as a potential arbovirus vector during inter-endemic periods, sustaining virus transmission for longer durations [49].

Regarding natural infections, the rate of DENV infection observed in *Ae. aegypti* was similar to that previously reported for some endemic in Colombia [50]. Other studies screening pools of female *Ae. aegypti* for DENV, ZIKV, and CHIKV have reported positive percentages ranging between 0–40.7% for DENV and 0–58.8% for CHIKV [25,28,50–52]. For DENV and CHIKV, we found 12.4% of *Ae. aegypti* females pools infected by DENV and CHIKV, while 41.2% of *Ae. albopictus* females were infected by CHIKV, which could highlight the efficient vector competence reported for certain *Ae. albopictus* populations and CHIKV strains [11]. *Ae. albopictus* was also found to be associated with considerable vertical and horizontal transmission of DENV [49]. CHIKV viruses also use more than one mosquito species as vectors to facilitate its spread in Brazil [53].

The DENV infection rate in *Ae. aegypti* is an acceptable predictor of dengue cases [50] and during our sampling period, 73 DENV-positive cases were reported in 2021, with 49 in Piamonte, 15 in Patía, and nine in Popayán. In 2022, 53 DENV cases in Piamonte, 12 in Patía, and four in Popayán have been reported [54], indicating concomitant arbovirus circulation in the study area; however, owing to the weaknesses of the case reporting system, distinguishing between the urban or rural origin of the cases and correlating the cases with entomo-virological surveillance were impossible. However, the presence of arboviruses in mosquitoes above 1800 masl and in rural areas are indications of arbovirus circulation and the potential impact on human health. Furthermore, in 2024, Popayán is facing the highest outbreak of DENV in its history, with 3,045 cases in the 30[th] week [54] suggesting the efficiency of *Ae. aegypti* and *Ae. albopictus* as vectors for DENV despite their low rate of contact with humans, as observed by the pupae per person and female per person indices [32]; as this is an important predictor for transmission as part of arbovirus R0 [55], however, they could be associated with high presence of DENV in Popayán is their other parameters are high enough.

Overall, our results demonstrate that integrating environmental, virological, and anthropogenic variables is crucial for the eco-epidemiological characterization of Aedes-borne viruses, especially in areas where two vector species with distinct ecological characteristics coexist. This approach contributes to a broader understanding of the geographic and temporal dynamics of these arboviruses.

## Supporting information

**S1 Fig.  Biplot of principal component analysis.** The best-represented variables at each component were selected according to the eigenvalues. The points represent sampled houses at urban (yellow triangles) and rural (blue points).
(TIFF)

**S1 Table.  Summarized data of urban and rural areas per sampled municipality.**
(DOCX)

**S2 Table.  Summarized information of water containers inspected in urban and rural areas, emphasizing positive breeding places for *Ae. aegypti* and *Ae. albopictus.***
(DOCX)

## Acknowledgments

We want to thank the administrative staff of Secretaría de Salud del Cauca, specially to Dr. Duban Quintero and Dr. Anderson Piamba. To the Piamonte, Patía, Piendamó, and Popayán mayors' office, and to their local health secretaries for their logistic support during the fieldwork. We want to thank specially to our fieldworkers Erica Pame, Elvesio Narváez, Victor Perdomo, Cristian Alegría, and Jarvis López. To the local community from urban and rural areas of Piamonte, Patía, Piendamó, and Popayán for receiving us and allowing us to carry out our research. We especially want to thank Dr. Victor Alberto Olano for his advice and expertise during the conception of the original research proposal. We would like to thank Editage (www.editage.com) for English language editing.

## Author contributions

**Conceptualization:** Juan S. Mantilla-Granados, Karol Montilla-López, Diana Sarmiento-Senior, Myriam Lucía Velandia-Romero, Carlos Andrés Morales, Jaime E. Castellanos.

**Data curation:** Juan S. Mantilla-Granados, Karol Montilla-López, Diana Sarmiento-Senior, Eliana Calvo.

**Formal analysis:** Juan S. Mantilla-Granados, Karol Montilla-López, Diana Sarmiento-Senior, Elver Chapal-Arcos, Myriam Lucía Velandia-Romero, Eliana Calvo, Jaime E. Castellanos.

**Funding acquisition:** Juan S. Mantilla-Granados, Diana Sarmiento-Senior, Myriam Lucía Velandia-Romero, Carlos Andrés Morales, Jaime E. Castellanos.

**Investigation:** Juan S. Mantilla-Granados, Karol Montilla-López, Diana Sarmiento-Senior, Elver Chapal-Arcos, Myriam Lucía Velandia-Romero, Eliana Calvo, Carlos Andrés Morales, Jaime E. Castellanos.

**Methodology:** Juan S. Mantilla-Granados, Karol Montilla-López, Diana Sarmiento-Senior, Elver Chapal-Arcos, Myriam Lucía Velandia-Romero, Eliana Calvo, Carlos Andrés Morales, Jaime E. Castellanos.

**Project administration:** Juan S. Mantilla-Granados, Myriam Lucía Velandia-Romero, Jaime E. Castellanos.

**Resources:** Juan S. Mantilla-Granados, Eliana Calvo, Carlos Andrés Morales, Jaime E. Castellanos.

**Software:** Juan S. Mantilla-Granados.

**Supervision:** Juan S. Mantilla-Granados, Diana Sarmiento-Senior, Myriam Lucía Velandia-Romero, Eliana Calvo, Carlos Andrés Morales, Jaime E. Castellanos.

**Validation:** Juan S. Mantilla, Eliana Calvo.

**Visualization:** Juan S. Mantilla-Granados.

**Writing – original draft:** Juan S. Mantilla-Granados, Karol Montilla-López, Myriam Lucía Velandia-Romero, Jaime E. Castellanos.

**Writing – review & editing:** Juan S. Mantilla-Granados, Karol Montilla-López, Diana Sarmiento-Senior, Elver Chapal-Arcos, Myriam Lucía Velandia-Romero, Eliana Calvo, Carlos Andrés Morales, Jaime E. Castellanos.

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
