## [Decision Letter · Decision Letter 0]

6 Jan 2025

PNTD-D-24-01455Environmental and anthropic factors influencing Aedes aegypti and Aedes albopictus (Diptera: Culicidae), with emphasis on natural infection and dissemination: Implications for an emerging vector in Colombia.PLOS Neglected Tropical Diseases Dear Dr. Mantilla, Thank you for submitting your manuscript to PLOS Neglected Tropical Diseases. After careful consideration, we feel that it has merit but does not fully meet PLOS Neglected Tropical Diseases's publication criteria as it currently stands. Therefore, we invite you to submit a revised version of the manuscript that addresses the points raised during the review process. Please submit your revised manuscript within 30 days Mar 07 2025 11:59PM. If you will need more time than this to complete your revisions, please reply to this message or contact the journal office at plosntds@plos.org. Please include the following items when submitting your revised manuscript: * A rebuttal letter that responds to each point raised by the editor and reviewer(s). You should upload this letter as a separate file labeled 'Response to Reviewers '. This file does not need to include responses to any formatting updates and technical items listed in the 'Journal Requirements' section below. * A marked-up copy of your manuscript that highlights changes made to the original version. You should upload this as a separate file labeled 'Revised Manuscript with Track Changes ' * An unmarked version of your revised paper without tracked changes. You should upload this as a separate file labeled 'Manuscript '. If you would like to make changes to your financial disclosure, competing interests statement, or data availability statement, please make these updates within the submission form at the time of resubmission. Guidelines for resubmitting your figure files are available below the reviewer comments at the end of this letter. We look forward to receiving your revised manuscript. Kind regards, Richard A. BowenAcademic EditorPLOS Neglected Tropical Diseases Victoria BrookesSection EditorPLOS Neglected Tropical Diseases

Shaden Kamhawi

co-Editor-in-Chief

Paul Brindley

co-Editor-in-Chief

**Journal Requirements:**

1) We do not publish any copyright or trademark symbols that usually accompany proprietary names, eg ©,  ®, or TM  (e.g. next to drug or reagent names). Therefore please remove all instances of trademark/copyright symbols throughout the text, including:

- ® on page: 11 line 220 and line 222.

- TM on page: 8 line 151 and line 153.

3) The file inventory includes files for Figures 3a, and 3b. We would recommend either combining these into a single Figure 3.tiff file with separate internal panels, or renumbering them as individual figures, and as we are not able to publish multiple components of a single figure as separate files.

4) Tables should not be uploaded as individual files. Please remove these files and include the Tables in your manuscript file as editable, cell-based objects. For more information about how to format tables, see our guidelines:

https://journals.plos.org/plosntds/s/tables 

5) We have noticed that you have uploaded Supporting Information files, but you have not included a complete list of legends. Please add a full list of legends including (Paper data file )for your Supporting Information files after the references list.

6) We notice that your supplementary Figure, and Table are included in the manuscript file. Please remove them and upload them with the file type 'Supporting Information'. Please ensure that each Supporting Information file has a legend listed in the manuscript after the references list.

7) Some material included in your submission may be copyrighted. According to PLOSu2019s copyright policy, authors who use figures or other material (e.g., graphics, clipart, maps) from another author or copyright holder must demonstrate or obtain permission to publish this material under the Creative Commons Attribution 4.0 International (CC BY 4.0) License used by PLOS journals. Please closely review the details of PLOSu2019s copyright requirements here: PLOS Licenses and Copyright. If you need to request permissions from a copyright holder, you may use PLOS's Copyright Content Permission form.

Potential Copyright Issues:

i) Figure 1. Please (a) provide a direct link to the base layer of the map (i.e., the country or region border shape) and ensure this is also included in the figure legend; and (b) provide a link to the terms of use / license information for the base layer image or shapefile. We cannot publish proprietary or copyrighted maps (e.g. Google Maps, Mapquest) and the terms of use for your map base layer must be compatible with our CC BY 4.0 license.

8) We note that your Data Availability Statement is currently as follows: "The data supporting the findings of this study have been uploaded as a .txt file and are publicly available." Please confirm at this time whether or not your submission contains all raw data required to replicate the results of your study. Authors must share the “minimal data set” for their submission. PLOS defines the minimal data set to consist of the data required to replicate all study findings reported in the article, as well as related metadata and methods (https://journals.plos.org/plosone/s/data-availability#loc-minimal-data-set-definition).

9) Please amend your detailed Financial Disclosure statement. This is published with the article. It must therefore be completed in full sentences and contain the exact wording you wish to be published.

**Reviewers' comments:** Reviewer's Responses to Questions

**Key Review Criteria Required for Acceptance?**

**Methods**

-Are the objectives of the study clearly articulated with a clear testable hypothesis stated?

-Is the study design appropriate to address the stated objectives?

-Is the population clearly described and appropriate for the hypothesis being tested?

-Is the sample size sufficient to ensure adequate power to address the hypothesis being tested?

-Were correct statistical analysis used to support conclusions?

-Are there concerns about ethical or regulatory requirements being met?

Reviewer #1: The objetives are well achieved. The design study was appropriate to achieve the objectives. The vector population is correctly described and accomplished. The sample and the statistical analysis were adequate to reach the conclusions. There are not any concerns about ethical and regulatory requirements.

Reviewer #2: I do think that objectives and methods are clearly articulated. However, some improvements of the writing should be done in order to fully understand them. Some of my main concerns are the following:

I understand that localities were chosen by consider their elevation and consequently their temperature range differs. However, how is their endemicity status? I think authors should include information about this in the description of the study area as it can bias the infection results.

How were sampled households selected? Was a randomizing software used?

I suggest improving the writing so the categorization into artificial and natural containers in clearer.

Please specify what Zeiss ™ Stemi ™ DV4 is. Is it a stereo microscope?

What period of time was used to collect climatic data from WorldClim? It’s important to mention if the climatic data corresponds well with the time of collection. Otherwise, why was such data used?

Explain further how the principal components analysis was done? The vast majority of the section “Anthropogenic variables” are results. I would strongly suggest to rewrite this section and leave only methods here and move the remaining to Results.

It is not clear to me how authors estimate the risk of mosquito biting from density of females data. I strongly suggest to explain this beyond the inclusion of a reference.

Are in these areas circulating mosquito species other than the two mentioned in the manuscript? E.g. culex species? If so, how authors clean the data related to residents’ responses about their knowledge on Aedes from those of other species?

Reviewer #3: (No Response)

**Results**

-Does the analysis presented match the analysis plan?

-Are the results clearly and completely presented?

-Are the figures (Tables, Images) of sufficient quality for clarity?

Reviewer #1: Yes, analysis is in accordance with analysis plan. Results are well described and presented. Figures and tables are also well showed and displayed.

Reviewer #2: The results do match the analysis plan. I found them though extremely detailed and it seems to me that authors should do an extra effort to improve description so they can focus better on the important results.

I have other minor comments to fully understand results:

Please indicate how many houses per locality were visited twice and how are authors handling the data so these houses are not biasing the analysis.

I am not sure how the biting risk was estimated. Without such knowledge, those results are not biologically meaningful.

Was exactly the same percentage of pools positive for both DENV and CHIKV or is it a typo?

It seems a little odd that ZIKV was found in males and not in the 14 evaluated female pools. Was this result re-examined to confirm that no cross-reaction or contamination happened?

Reviewer #3: (No Response)

**Conclusions**

-Are the conclusions supported by the data presented?

-Are the limitations of analysis clearly described?

-Do the authors discuss how these data can be helpful to advance our understanding of the topic under study?

-Is public health relevance addressed?

Reviewer #1: Conclusions are suppported by data presented but limitations of analysis were not considered that meaning that the authors did not find any restrictions to reach the conclusions. They discussed data can be relevant to the control of Aedes vectors in public health.

Reviewer #2: The discussion could be improved to help readers more easily identify broader patterns. While the results do support the conclusions, I found myself having to revisit sections of the manuscript multiple times to fully understand them.

Reviewer #3: (No Response)

**Editorial and Data Presentation Modifications?**

Reviewer #1: There is a list of abbreviation for generic and subgeneric taxa in family Culicidae by Reinert which is highly recommended to follow.

I suggest a minor revision considering this issue.

Reviewer #2: Figure 1: I suggest including additional maps where each locality is shown separately. In its current way, the sampled households (points) are not correctly visualized.

Line 196: change cero by zero

Consider rewrite line 205-206.

What authors means by “high house conditions” and “lowest house conditions”?

line 477: maybe the authors might indicate R0 (with zero: Basic reproduction number)? Is different from Ro.

Reviewer #3: Major comments:

- A albopictus is already an established vector for numerous arboviruses, including all those discussed in this manuscript (DENV, CHIKV, ZIKV). While this fact is acknowledged in the paper, the authors should make it more clear as to why there is a question regarding A albopictus being a vector IN COLUMBIA.

- Table 2 is not helpful; it should be either moved to supplemental material if kept in its current format or else replaced with a figure or simplified table to more clearly present the data.

- The data regarding public knowledge of mosquito and arboviral characteristics seems out of place and not in line with the general focus of the manuscript and contributes very little, if any, to the overall story here.

- It is mentioned numerous times in the manuscript that A albopictus showed evidence of "disseminated" infection (virus in legs, salivary glands, etc.), whereas no such mention is made regarding A aegypti. Presumably this is because (1) A aegypti were processed in batches and (2) A aegypti is already the accepted primary vector. Whatever the case, this should be clearly explained. Table 4 is relevant here, as it apparently is referencing only A albopictus; the title of the table should reflect this rather than provide the generic "Arbovirus dissemination".

Minor comments:

- "masl" should be specified as the abbreviation for "meters above sea level" in the abstract.

- Line 31-32: a rational should be provided as to why different testing methods were utilized for the different species of mosquitoes.

- Line 39: a unit should be provided for the numbers in parentheses (presumably females per person).

- It is unclear why there are two shades of aquamarine in Figure 1 (visible in northwest corner of inset).

- The speculative comments about Mayaro virus (lines 479-485) should be removed.

- There are numerous typographical and grammatical errors throughout the manuscript.

**Summary and General Comments**

Reviewer #1: The authors sampled an equivalent number of houses in the two zones of Cauca. There was not information about the general climate and the mountainuos topography of Cauca. In what extension does the topography defines the urban and rural zones? Is there a gradient of human occupancy in the two zones? These questions are made to clearly distinguish the boundaries if they exist and describe if there is any contact between the two zones. The proximity of houses in the two zones would improve the distribution of the two vectors if there is any contact. This issue is very important to characterize the proximity of breeding places in the zones and should be considered in the analysis.

Reviewer #2: The study provides an interesting analysis of ecological and socio-demographic variables influencing the presence of arbovirus vectors and their infection. However, the manuscript is highly descriptive and appears to focus on very localized findings. To make the study more appealing to a broader audience, I suggest keeping the manuscript as concise and informative as possible, minimizing overly detailed descriptions of specific results. At times, it was difficult to identify broader patterns that would enhance the paper's general relevance.

That said, I believe the study is suitable for publication, but I think some improvements are needed before its release.

Reviewer #3: The following sentences seem to be the best summary of the paper in the author's own words:

- lines 99-103, "In Colombia, A. albopictus was first reported in 1998, and has since spread to 32 departments of the country. This mosquito is naturally infected by DENV, ZIKV, and CHIKV with some studies demonstrating its ability to colonize major urban areas. However, the role of this species as a vector in Colombia, its relationship with A. aegypti, and the climatic and anthropogenic factors associated with their distribution are poorly understood."

- lines 390-392, "Our detailed and integrative analysis of ecological, environmental, and sociodemographic factors, combined with molecular detection and mosquito dissections, helped us establish detailed information about the ecoepidemiology of Aedes-borne viruses in the study area."

The manuscript should be structured around such statements to better clarify the purpose of the study and the actual findings that are present. It is very easy to get lost in the details of this manuscript, and it needs to flow and tell the story better overall. Furthermore, some of the other summary statements which make reference to the study "improving control strategies" or revealing "arbovirus transmission" patterns are overreaches and should be removed.

PLOS authors have the option to publish the peer review history of their article (what does this mean? ). If published, this will include your full peer review and any attached files.

**Do you want your identity to be public for this peer review?** For information about this choice, including consent withdrawal, please see our Privacy Policy .

Reviewer #1: **Yes: ** Rosa Maria Tubaki

Reviewer #2: No

Reviewer #3: No

---

## [Editor Report · Decision Letter 1]

25 Mar 2025

Dear Mr Mantilla,

We are pleased to inform you that your manuscript 'Environmental and anthropic factors influencing Aedes aegypti and Aedes albopictus (Diptera: Culicidae), with emphasis on natural infection and dissemination: Implications for an emerging vector in Colombia.' has been provisionally accepted for publication in PLOS Neglected Tropical Diseases.

Best regards,

Richard Bowen

Academic Editor

Victoria Brookes

Section Editor

Shaden Kamhawi

co-Editor-in-Chief

Paul Brindley

co-Editor-in-Chief

---

## [Editor Report · Acceptance letter]

Dear Mr Mantilla,

We are delighted to inform you that your manuscript, "Environmental and anthropic factors influencing Aedes aegypti and Aedes albopictus (Diptera: Culicidae), with emphasis on natural infection and dissemination: Implications for an emerging vector in Colombia.," has been formally accepted for publication in PLOS Neglected Tropical Diseases.

Best regards,

Shaden Kamhawi

co-Editor-in-Chief

Paul Brindley

co-Editor-in-Chief
